# Multimodal principal component analysis to identify major features of white matter structure and links to reading

Bryce L. Geeraert[1,2]*, Maxime Chamberland[3], R. Marc Lebel[2,4,5], Catherine Lebel[2,4]

**1** Biomedical Engineering Graduate Program, University of Calgary, Calgary, Alberta, Canada, **2** Alberta Children's Hospital Research Institute, University of Calgary, Calgary, Alberta, Canada, **3** School of Psychology, Cardiff University Brain Research Imaging Centre (CUBRIC), Cardiff, United Kingdom, **4** Department of Radiology, University of Calgary, Calgary, Alberta, Canada, **5** GE Healthcare, Calgary, Alberta, Canada

* blgeerae@ucalgary.ca

**Data Availability Statement:** Data will be held in a public repository on FigShare (https://figshare.com/). It is currently accessible at: https://doi.org/10.6084/m9.figshare.12649388.

## Abstract

The role of white matter in reading has been established by diffusion tensor imaging (DTI), but DTI cannot identify specific microstructural features driving these relationships. Neurite orientation dispersion and density imaging (NODDI), inhomogeneous magnetization transfer (ihMT) and multicomponent driven equilibrium single-pulse observation of T1/T2 (mcDESPOT) can be used to link more specific aspects of white matter microstructure and reading due to their sensitivity to axonal packing and fiber coherence (NODDI) and myelin (ihMT and mcDESPOT). We applied principal component analysis (PCA) to combine DTI, NODDI, ihMT and mcDESPOT measures (10 in total), identify major features of white matter structure, and link these features to both reading and age. Analysis was performed for nine reading-related tracts in 46 neurotypical 6–16 year olds. We identified three principal components (PCs) which explained 79.5% of variance in our dataset. PC1 probed tissue complexity, PC2 described myelin and axonal packing, while PC3 was related to axonal diameter. Mixed effects regression models did not identify any significant relationships between principal components and reading skill. Bayes factor analysis revealed that the absence of relationships was not due to low power. Increasing PC1 in the left arcuate fasciculus with age suggest increases in tissue complexity, while increases of PC2 in the bilateral arcuate, inferior longitudinal, inferior fronto-occipital fasciculi, and splenium suggest increases in myelin and axonal packing with age. Multimodal white matter imaging and PCA provide microstructurally informative, powerful principal components which can be used by future studies of development and cognition. Our findings suggest major features of white matter undergo development during childhood and adolescence, but changes are not linked to reading during this period in our typically-developing sample.

## Introduction

Reading is a sophisticated skill with many constituent systems including vision, language, memory, and attention. White matter fibers play an important role in connecting these

**Funding:** Author RML is an employee of GE Healthcare. The funder provided support in the form of salaries for author RML, but did not have any additional role in the study design, data collection and analysis, decision to publish, or preparation of them anuscript. The specific roles of these authors are articulated in the 'author contributions' section.

**Competing interests:** I have read the journal's policy and the authors of this manuscript have the following competing interests: Author RML is an employee of GE Healthcare. The funder provided support in the form of salaries for author RML, but played no other role in this study. This does not alter our adherence to PLOS ONE policies on sharing data and materials.

systems and facilitating coordinated processing across the reading network. Diffusion tensor imaging (DTI) is frequently used to investigate links between white matter and reading thanks to its sensitivity to white matter microstructural features. DTI studies have linked reading to white matter in a broad network of tracts including the arcuate, superior and inferior longitudinal, inferior fronto-occipital, and uncinate fasciculi, and the posterior corpus callosum [1–5], such that markers of increased white matter maturity correlate with better reading scores. Additionally, longitudinal DTI studies show that maturation of reading-related tracts is related to improvements in reading ability [6–10]. White matter abnormalities have been observed in children with reading difficulties, most often in left temporo-parietal white matter [11–14] as language and reading networks are typically left lateralized [11, 15, 16]. Finally, changes in DTI measures are observed in reading-related white matter following reading interventions [17–19].

DTI studies have identified a network of white matter related to reading but cannot comment on the particular features of white matter microstructure driving these relationships. Fractional anisotropy (FA) and mean diffusivity (MD) describe water diffusion and are simultaneously sensitive to many microstructural factors [20–23]. Newer techniques with increased specificity may be used to build upon DTI literature. Neurite orientation dispersion and density imaging (NODDI) produces the neurite density index (NDI) and orientation dispersion index (ODI) which are sensitive to axonal packing and tract coherence, respectively [24]. Inhomogeneous magnetization transfer (ihMT) and multicomponent driven equilibrium single-pulse observation of T1 and T2 (mcDESPOT) produce the quantitative ihMT (qihMT) and myelin volume fraction ($VF_m$) measures respectively, both sensitive to myelin [25, 26]. Additionally, measures of axon volume and myelin volume such as NDI and $VF_m$ can be combined to produce the g-ratio, which describes the ratio of axon thickness to total fiber diameter [27]. These methods have been validated *in vitro* [28–33], and they hold great potential to clarify our understanding of white matter development and links to reading.

Investigating multiple imaging measures in a univariate fashion, the typical practice in developmental studies to date, necessarily increases the number of comparisons and may introduce redundancy via shared sensitivities between metrics, reducing the discriminating power of the analysis. One solution to reduce comparisons and exploit shared sensitivities is to collapse white matter measures into orthogonal components via principal component analysis (PCA). A framework using PCA for dimensionality reduction in white matter has been recently described [34], and resultant components were linked to age, suggesting developmental sensitivity. The goal of this study was to combine white matter imaging techniques (DTI, NODDI, ihMT, and mcDESPOT) to better understand relationships between brain structure and reading in a sample of healthy 6–16 year old children. We aimed to investigate links between resultant principal components and both age and reading to describe development of key microstructural features and how these features underlie reading. We hypothesized that observed principal components would represent diffusion restriction and tissue complexity, similar to previous studies [34]. Furthermore, we expected that these components would be linked to age and reading proficiency in reading-related tracts, such that indications of more myelin, axonal packing, and fiber coherence would increase with age and would relate to better reading performance.

## Methods

### 2.1 Participants

46 healthy participants aged 6–16 years (mean age: 11.0 ± 2.6 years, 24 males / 22 females) were recruited as part of an ongoing study on pediatric brain development. Inclusion criteria

were: 1) uncomplicated birth between 37–42 weeks' gestation, 2) no history of developmental disorder, psychiatric disease, or reading difficulty, 3) no history of neurosurgery, and 4) no contraindications to MRI. 22 children (mean age: 13.3 ± 2.6 years, 11 males / 11 females) returned 2 years after their initial visit for a second scan and cognitive assessment. All subjects provided informed assent and parents/guardians provided written informed consent. Gender was determined by parent report. This study was approved by the local research ethics board, Conjoint Health Research Ethics Board (CHREB, ID: REB13-1346). All subjects provided informed assent and parents/guardians provided written informed consent.

## 2.2 Imaging

Subjects were scanned using a 32-channel head coil on a GE 3T Discovery MR750w (GE, Milwaukee, WI) system at the Alberta Children's Hospital. Two diffusion-weighted datasets were sequentially acquired at b = 900 s/mm$^2$ and 2000 s/mm$^2$ using a spin-echo echo planar imaging sequence with TR/TE = 12s/88ms, 2.2 mm x 2.2 mm x 2.2 mm resolution, with 5 b = 0 s/mm$^2$ volumes and 30 gradient directions per volume, scan time was 7:12 min:sec per diffusion dataset. IhMT images used a 3D spoiled gradient (SPGR) sequence: TR/TE = 10.46ms/2.18ms, 2.2mm x 2.2 mm x 2.2 mm resolution, flip angle 8˚. The sequence included a 5ms Fermi pulse with peak B1 of 45 mG and 5kHz offset prior to each excitation. The MT condition cycled between positive offset (+5kHz), dual offset (±5kHz), negative offset (-5kHz), and dual offset. A 32˚ flip angle reference image with no MT pulse was acquired for quantification. Scan time for ihMT was 5:12 min:sec. For mcDESPOT, multi-flip angle 3D SPGR images (α = 3˚, 4˚, 5˚, 6˚, 7˚, 9˚, 13˚, and 18˚) were collected with TR/TE = 9.1ms/3.9ms, 1.7mm x 0.86mm x 1.7mm resolution. Then, inversion recovery SPGR (IR-SPGR) images were collected to correct for B$_1$ inhomogeneity using 5˚ α, TR/TE = 9.1ms/3.9ms, 2.29mm x 0.86mm x 3.4mm resolution. Finally, two multi-flip angle balanced steady-state free precession (bSSFP) images were collected at phase 0˚ and 180˚, with α = 10˚, 13˚, 16˚, 20˚, 23˚, 30˚, 43˚, and 60˚, TR/TE = 6.6ms/3.2ms, 1.7mm x 0.86mm x 1.7mm resolution. Collection of bSSFP images at two phases enables correction for B$_0$ inhomogeneity. Total scan time for all mcDESPOT scan sequences was 16:35 min:sec. T1-weighted anatomical images were also acquired, with TI = 600ms, TR/TE = 8.2ms/3.2ms, 0.8 mm x 0.8 mm x 0.8 mm resolution, scan time 5:38 min:sec.

## 2.3 Image processing

All images were visually inspected for quality assessment and processed separately using appropriate tools before being combined for principal component analysis. Preprocessing for T1 images was carried out in FreeSurfer 5.3 (http://surfer.nmr.mgh.harvard.edu/) for intensity normalization and brain extraction. Preprocessing for DTI datasets was performed within ExploreDTI [35]. Preprocessing steps included signal drift correction [36], brain extraction, eddy current and motion corrections [37, 38], and registration to skull-stripped T1 images to correct geometric distortions induced by echo-planar imaging. The REKINDLE model was used to calculate FA, MD, radial diffusivity (RD), and axial diffusivity (AD) maps for each subject using the b = 900 s/mm$^2$ shell only [39]. Whole brain tractography was performed on b = 900 s/mm$^2$ data using constrained spherical deconvolution [40] with L_max = 6, 2mm isotropic seed voxels, 1mm step size, FA threshold of 0.2, 30 maximum angle of deviation and an acceptable streamline range of 50 to 500mm. Following whole brain tractography, semiautomated methods [41] were performed to segment the arcuate, inferior longitudinal (ILF), inferior fronto-occipital (IFOF), and uncinate fasciculi bilaterally, along with the splenium, as shown in Fig 1. A 11-year old female with high data quality was selected as the exemplar participant for this process; all regions were drawn on this template brain and then registered to

other participants' data for tracking in native space [42]. Processed multi-shell DTI datasets were also exported to the NODDI Toolbox (http://www.nitrc.org/projects/noddi_toolbox) for calculation of isotropic ($f_{iso}$) and intracellular ($f_{icvf}$, or NDI) volume fractions and ODI.

Pseudo-quantitative ihMT maps (qihMT) and magnetization transfer ratio (MTR) maps were calculated from ihMT data using an in-house GE protocol as described in previous work [43]. Following MTR and qihMT image production, brain extraction was performed on MTR images using FSL's BET2 tool [44], and resulting brain-extracted MTR image was used as a mask to produce a brain-extracted qihMT image.

mcDESPOT SPGR, IR-SPGR, and bSSFP images were aligned to the SPGR image with the largest α then processed by fitting T1, T2, and volume fractions to three water compartments (myelin-bound, intra/extracellular, and free), along with exchange rates between myelin-bound and intra/extracellular water [45]. The myelin-bound water volume fraction from this fitting was used to produce $VF_m$ maps for each participant. G-ratio maps were computed using $VF_m$, NDI, and $f_{iso}$ maps to calculate the fiber volume fraction (FVF) and g-ratio using the following two equations.

$$FVF = VF_m + (1 - VF_m)(1 - f_{iso})NDI$$

$$g - ratio = \sqrt{(1 - VF_m)/FVF}$$

Following production of all measure maps, qihMT, MTR, $VF_m$, NDI, and ODI maps were registered to b = 900 s/mm$^2$ FA maps using Advanced Normalization Tools (ANTs) [46]. Default parameters from antsRegistrationSyN.sh were used, with the –t s flag chosen to select rigid, affine, and deformable symmetric normalization transforms. Then, the mean FA, MD, AD, RD, NDI, ODI, MTR, qihMT, $VF_m$, and g-ratio values were extracted for all 9 tracts of interest (Fig 1) per participant. Additionally, along-tract analysis was performed in Explore-eDTI [47, 48], to sample all ten measures at twenty equidistant points along each tract. Fig 2

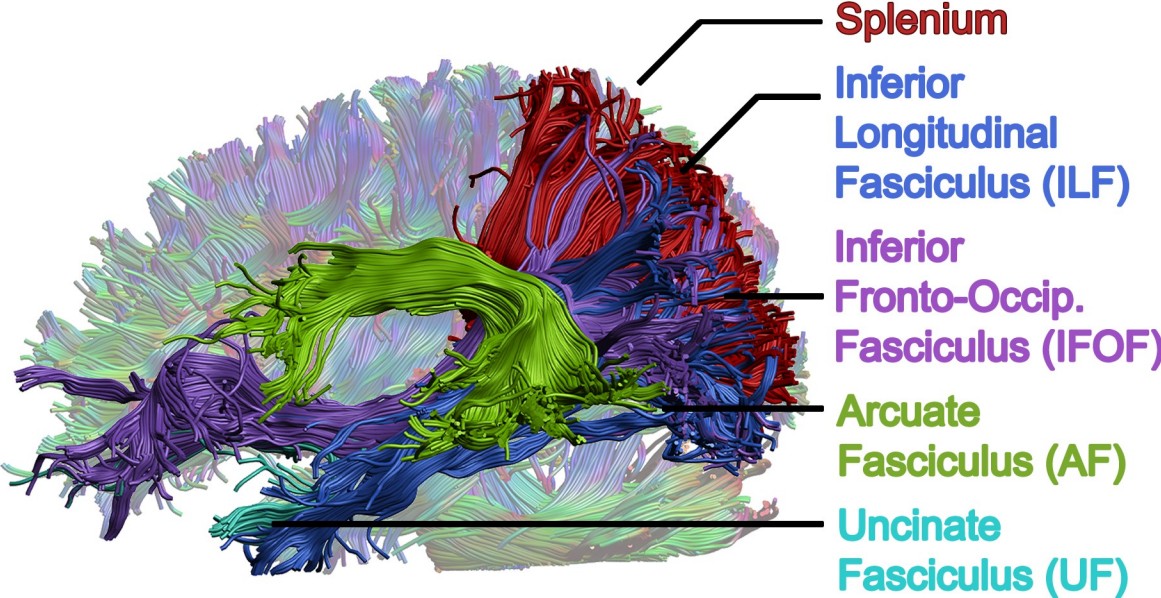

**Fig 1. Major, reading-related white matter tracts chosen as regions of interest.** Whole brain tractography was performed via constrained spherical deconvolution, then tracts were segmented using deterministic semi-automated methods in ExploreDTI. Regions of interest were investigated bilaterally, but only the left hemisphere is shown here.

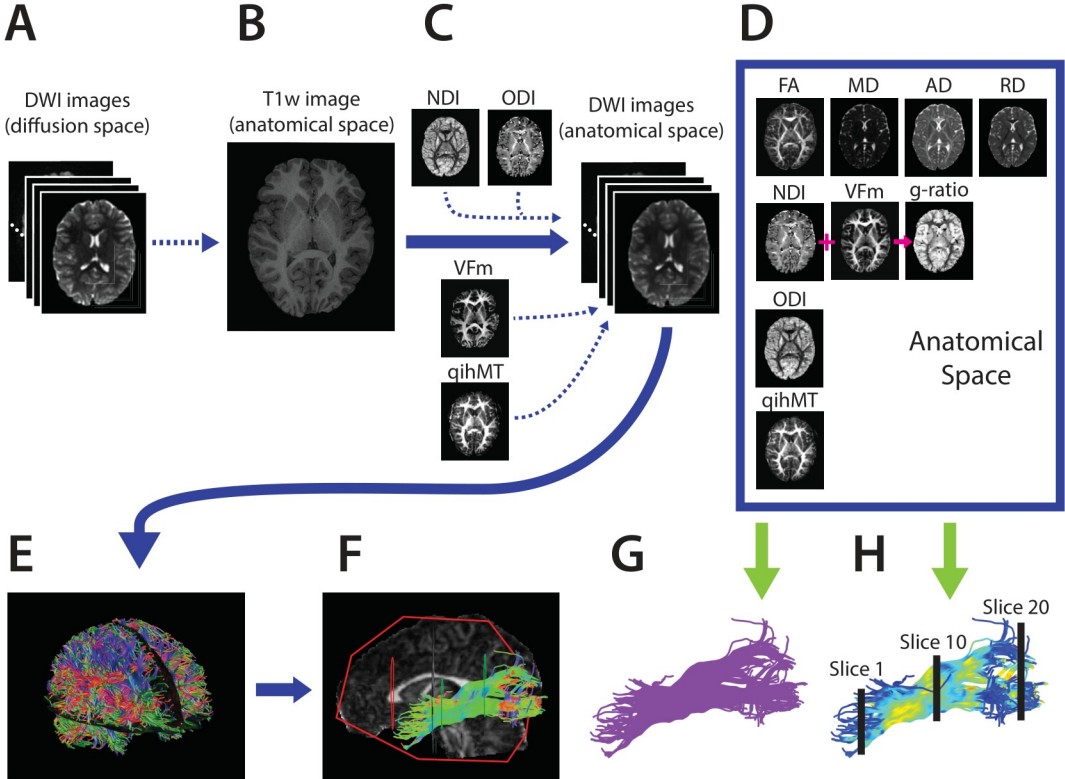

**Fig 2. Processing pipeline to prepare imaging data for principal component analysis.** Preprocessed diffusion-weighted images (A) were registered to T1-weighted anatomical images (B). Measure maps from NODDI, ihMT, and mcDESPOT sequences were registered to diffusion-weighted images in anatomical space (C) to produce all measure maps in anatomical space (D). Next, whole brain tractography was computed from b = 900s/mm² data using constrained spherical deconvolution (E), and tracts of interest were segmented in a semiautomated fashion in ExploreDTI (F). Measure means were extracted for each tract of interest (G) and along each tract of interest at 20 equidistant segments (H).

visually depicts all processing steps performed following preprocessing of images in their native space.

## 2.4 Reading assessments

Reading was evaluated using the Wechsler Individual Achievement Test–Third Edition: Canadian [49]. Participants completed the Reading Comprehension, Word Reading, Pseudoword Decoding, and Oral Reading Fluency subtests. From these subtests, the Total Reading Composite Score was computed as a measure of general reading proficiency. This score combines phonological awareness, reading comprehension, and fluency.

## 2.5 Principal component analysis

To implement principal component analysis in white matter, we followed the methods described in Chamberland et al [34]. All analysis was conducted in R version 3.6.1 [50]. First, along tract data for each subject's first time point (10 measures x 9 tracts of interest x 20 points along each tract) was combined into a single table for principal component analysis (described in Chamberland et al [34]). A Kaiser-Meyer-Olkin (KMO) test was conducted via the *KMO()* function to assess correlations between input measures and indicate the suitability of our

measure set for PCA; values >0.5 indicate suitability [51]. PCA was performed via the *prcomp ()* function (using the scale = 1 option to normalize each feature independently). Following PCA, input variable contributions to principal components along with correlations between variables within along-tract data were inspected to identify redundancy between variables. In the case of highly collinear measures (moderately to highly correlated ($|r| > 0.6$) and contributed to PCA outputs similarly), the variable with highest correlations to all other input measures was removed to improve stability of PCA computations [52] and PCA was recomputed. Principal components with eigenvalue > 1 were retained, while other components were discarded [53]. Varimax rotation was applied on retained principal components via the *varimax ()* function to maximize differences in principal components loadings and improve interpretation of component sensitivities. Measures were considered meaningful contributors to a resultant principal component if they accounted for above average variance (>11.1%) in the component.

## 2.6 Statistical analysis

All statistical analysis was performed in R version 3.6.1 [50]. Following varimax rotation, longitudinal principal component weightings were calculated by multiplication of time point 2 along tract data with the rotation matrix output by *varimax()*. Next, along tract weightings for principal components were averaged in each tract to produce mean principal component weightings for each subject in all 9 investigated tracts. Linear mixed effects models were computed via *lmer()* [54] to investigate relationships between principal components with Total Reading and age in each tract. Age models included age, gender, an age*gender interaction, and a random intercept per subject, to account for repeated measures within subjects. If the age*gender interaction was not significant, it was removed and the model was rerun. Total Reading models for each tract included all retained principal components along with age, and gender if a gender effect was observed for any principal component. Restricted maximum likelihood was used for all models. Benjamini-Hochberg false discovery rate (FDR) correction was used to correct for 27 comparisons (9 tracts x three principal components). Multiple comparisons corrections were conducted separately for age and Total Reading findings. Example formulas are provided below. Time point 1 data for each measure included in our final PCA was correlated with Total Reading via partial correlation in each region, controlling for age, and FDR correction was applied for 9 correlations across each measure.

$$PC1 \sim Age + Gender + Age*Gender + (1|Subject)$$

$$Total\ Reading \sim PC1 + PC2 + PC3 + Age + Gender + (1|Subject)$$

Bayes factor analysis was performed via *generalTestBF* in the BayesFactor package for R [55] to supplement regression analysis by assessing the observed statistical power of models connecting retained principal components and Total Reading. Bayes factors output by *generalTestBF* were inverted to reflect the ratio of likelihood of the null hypothesis divided by the likelihood of a given model. A Bayes factor greater than 3, indicating our data was 3 times more likely to be described by the null hypothesis than a given model, was considered evidence for the null hypothesis. A Bayes factor less than 1/3, indicating that a model including our chosen predictors was 3 times more likely to explain our data than the null hypothesis, was considered evidence for the alternative hypothesis. Bayes factors between 1/3 and 3 were considered indicators of low power, such that neither evidence for the null or alternative hypotheses could be inferred [56].

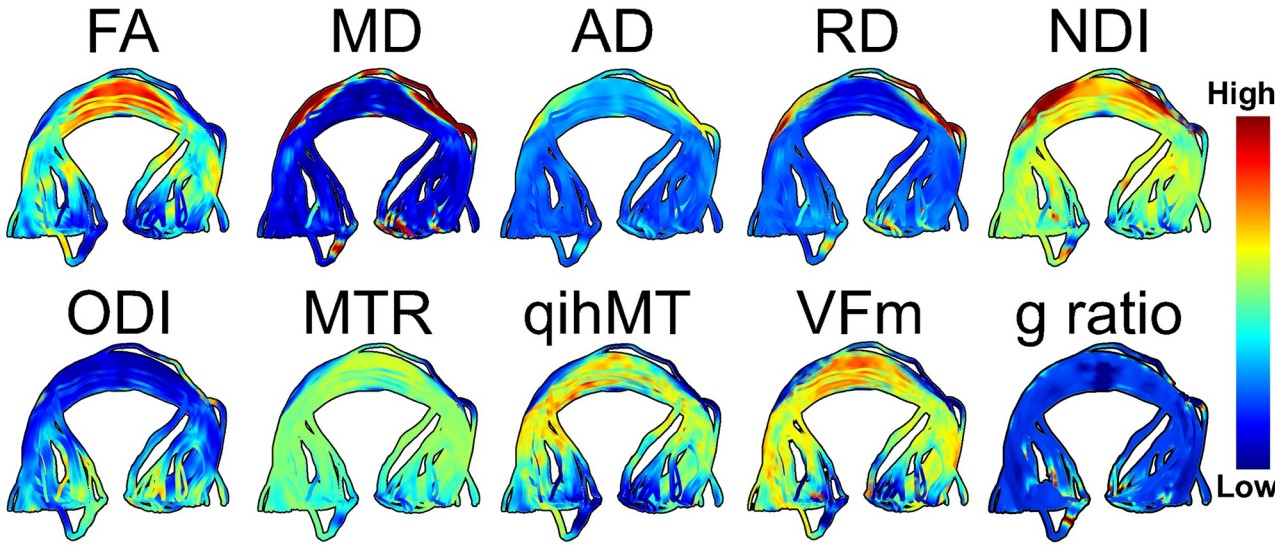

**Fig 3. Multimodal imaging of white matter microstructure in the splenium.** Measures from DTI, NODDI, MT, and mcDESPOT imaging can be contrasted to provide a multifaceted understanding of white matter structure.

## Results

### 3.1 Principal component analysis

Fig 3 visualizes each included imaging metric in the splenium. Here we can see that measures with shared sensitivities vary similarly across the tract. For example, FA, RD, qihMT, and $VF_m$ are all similar to myelin and reach extreme values in the center of the splenium (highly positive for FA, qihMT, and $VF_m$, highly negative for RD).

MTR was removed from our principal component analysis due to high collinearity with qihMT ($r^2$ = 0.64). Three principal components were identified in our final model, which collectively explained 79.5% of variance (KMO test value = 0.53). Measures contributing greater than 11.1% variance (expected if all variables contributed uniformly) to a component following varimax rotation are visualized in Fig 4. Interpretation of principal components was carried out by evaluating the common microstructural sensitivities of each measure, and by comparison to previous PCA analyses in white matter [34, 57]. Principal component (PC) 1 explained 37.5% of variance and was primarily composed of measures sensitive to tissue complexity: FA, AD, ODI, along with MD. PC2 explained 23.0% of variance and was composed of measures sensitive to myelin and axon packing: FA, MD, RD, and NDI. PC3 explained 19.0% of variance and was driven by measures sensitive to myelin and axonal diameter, $VF_m$ and g-ratio.

As shown in Fig 4 panel A, FA and MD contributed strongly to PC1 and PC2 even after varimax rotation, likely because FA and MD are broadly sensitive to white matter structure. To better interpret components, we removed FA and MD and recomputed PCA (results shown in Fig 4, panel B). The reduced model (denoted as $PC_B$) had three principal components that explained 77.3% of variance (KMO = 0.43). $PC1_B$ explained 36.6% of variance and was composed of RD, NDI, and qihMT. $PC2_B$ explained 22.7% of variance and was composed of VFm and g-ratio. Finally, $PC3_B$ explained 18.0% of variance and was driven by AD and ODI. Mixed effects regression models and Bayes factor analyses were conducted with the full PCA model including FA and MD to provide comparable data to previous studies, to preserve power to detect age and reading effects, and because the KMO test value of 0.43 for $PC_B$ indicated that input variables may not share enough information for robust factor analysis.

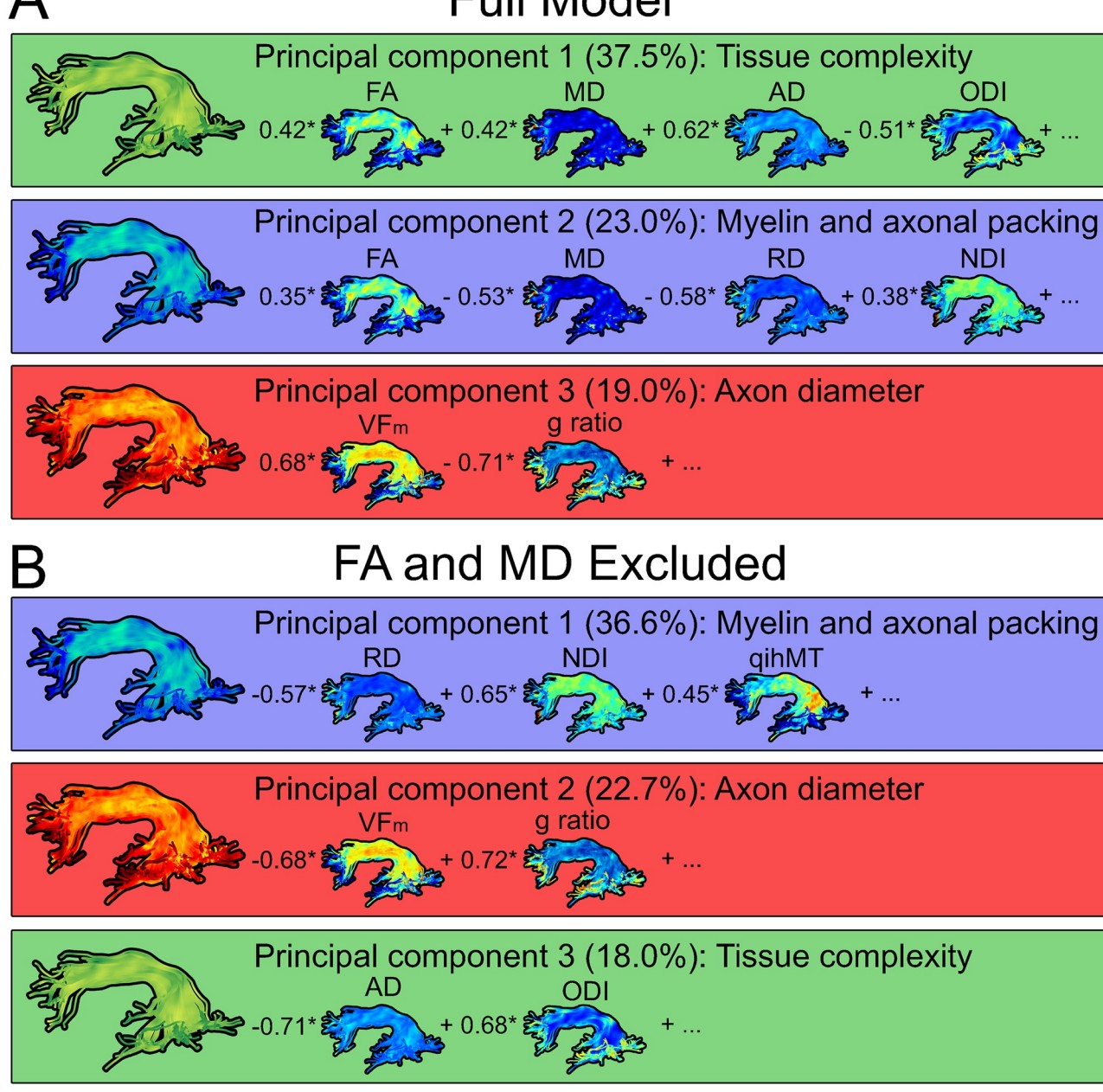

**Fig 4. Principal components visualized in the left arcuate fasciculus.** Correlations for measures which contribute greater variance than expected by chance (>11.1%) are included for each component. Panel A displays PCA results from all 9 measures. Components in Panel A explained 79.5% of variance in our data (variance explained by each individual component is noted in brackets). Principal components were related to diffusion along a primary axis (PC1), myelin and axonal packing (PC2), and axon diameter (PC3). Panel B shows results from a secondary PCA with FA and MD removed, as they loaded onto multiple components. Principal components in Panel B explain 77.3% of variance.

### 3.2 Regression models

Mixed effects models results linking principal components to Total Reading scores are summarized in Table 1. No significant relationships were observed between principal components and Total Reading. To further investigate the absence of significant relationships between

**Table 1. Parameters for mixed effects models linking principal components to Total Reading (formula: *Total Reading ~ PC1 + PC2 + PC3 + Age + (1|Subject)*).**

| Region | $R^2$ (adj) | df | Predictor | Estimate ± SE | t | p |
|---|---|---|---|---|---|---|
| Left arcuate | 0.026 | 64 | PC1 | -1.17 ± 6.61 | -0.18 | 0.860 |
| | | | PC2 | 5.26 ± 3.86 | 1.37 | 0.178 |
| | | | PC3 | 1.52 ± 2.89 | 0.53 | 0.601 |
| | | | Age | -0.00 ± 0.00 | -0.74 | 0.462 |
| Right arcuate | 0.022 | 64 | PC1 | -6.55 ± 5.42 | -1.21 | 0.232 |
| | | | PC2 | -0.13 ± 3.47 | -0.04 | 0.970 |
| | | | PC3 | -4.06E-2 ± 2.47 | -0.02 | 0.987 |
| | | | Age | -8.06E-4 ± 1.72E-3 | -0.47 | 0.641 |
| Left ILF | 0.026 | 64 | PC1 | -5.19 ± 6.44 | -0.81 | 0.424 |
| | | | PC2 | -0.88 ± 4.06 | -0.22 | 0.829 |
| | | | PC3 | -3.34 ± 2.60 | -1.28 | 0.207 |
| | | | Age | 6.64E-5 ± 1.62E-3 | 0.04 | 0.968 |
| Right ILF | 0.034 | 64 | PC1 | 6.06 ± 5.79 | 1.05 | 0.300 |
| | | | PC2 | 3.12 ± 3.87 | 0.81 | 0.423 |
| | | | PC3 | 0.89 ± 2.30 | 0.39 | 0.701 |
| | | | Age | -7.32E-4 ± 1.64E-3 | -0.45 | 0.656 |
| | | | Gender | 1.37 ± 3.74 | 0.37 | 0.716 |
| Left IFOF | 0.052 | 64 | PC1 | 0.75 ± 5.74 | 0.13 | 0.897 |
| | | | PC2 | 8.87 ± 5.01 | 1.77 | 0.081 |
| | | | PC3 | -1.43 ± 2.61 | -0.55 | 0.585 |
| | | | Age | -0.00 ±0.00 | -0.79 | 0.435 |
| Right IFOF | 0.046 | 64 | PC1 | -1.81 ±5.94 | -0.30 | 0.762 |
| | | | PC2 | 6.47 ± 3.99 | 1.62 | 0.110 |
| | | | PC3 | 2.08 ± 2.65 | 0.78 | 0.436 |
| | | | Age | -0.00 ± 0.00 | -0.93 | 0.356 |
| Left uncinate | 0.087 | 64 | PC1 | -5.83 ± 5.43 | -1.08 | 0.287 |
| | | | PC2 | 5.72 ± 4.34 | 1.32 | 0.192 |
| | | | PC3 | -3.85 ± 1.94 | -1.99 | 0.053 |
| | | | Age | -3.78E-4 ± 1.59E-3 | -0.24 | 0.813 |
| Right uncinate | 0.008 | 64 | PC1 | 1.16 ± 6.18 | 0.19 | 0.852 |
| | | | PC2 | 2.54 ± 3.38 | 0.75 | 0.455 |
| | | | PC3 | -0.49 ± 2.45 | -0.20 | 0.844 |
| | | | Age | -1.83E-4 ± 1.59E-3 | -0.12 | 0.909 |
| Splenium | 0.035 | 64 | PC1 | 7.28 ± 4.55 | 1.60 | 0.115 |
| | | | PC2 | 2.07 ± 2.50 | 0.83 | 0.410 |
| | | | PC3 | 1.45 ± 2.77 | 0.52 | 0.603 |
| | | | Age | -0.00 ± 0.00 | -0.28 | 0.778 |

principal components and Total Reading, we followed up by running mixed effects models between principal components and subtest scores for Reading Comprehension, Word Reading, Pseudoword Decoding, and Oral Reading Fluency. No significant relationships were observed between principal components and reading subtest scores. Correlations between the initial measure set and Total Reading are summarized in S1 Table. No significant correlations were observed between individual measures and Total Reading scores.

Table 2 summarizes models linking principal components to age and gender. A significant relationship between PC1 and age was observed in the left arcuate (t = -2.93, p = 0.004). Increases in PC1 with age suggest increased diffusion restrictions and tissue complexity

**Table 2. Parameters for mixed effects regression models linking principal components to age and gender (formula: *PC ~ Age + Gender+ Age\* Gender + (1|Subject)*).**

**PC1: Tissue Complexity**

| Region | R² (adj) | df | Predictor | Estimate ± SE | t | p |
|---|---|---|---|---|---|---|
| Left arcuate | 0.141 | 66 | Age | **8.71E-5 ± 2.92E-5** | **2.98** | **0.004**\* |
| | | | Gender | 7.97E-2 ± 6.6E-2 | 1.21 | 0.234 |
| Right arcuate | 0.091 | 66 | Age | 7.80E-5 ± 3.55E-5 | 2.20 | 0.032 |
| | | | Gender | 9.20E-2 ± 7.90E-2 | 1.16 | 0.251 |
| Left ILF | 0.005 | 66 | Age | 1.83E-5 ± 3.21E-5 | 0.57 | 0.571 |
| | | | Gender | -1.29E-2 ± 7.81E-2 | -0.17 | 0.870 |
| Right ILF | 0.031 | 65 | Age | 6.40E-6 ± 3.35E-5 | 0.19 | 0.849 |
| | | | Gender | -5.56E-2 ± 8.40E-2 | -0.66 | 0.513 |
| Left IFOF | 7.81E-5 | 66 | Age | 2.41E-6 ± 3.35E-5 | 0.07 | 0.943 |
| | | | Gender | 6.89E-4 ± 7.89E-2 | 0.01 | 0.993 |
| Right IFOF | 0.025 | 66 | Age | 4.14E-5 ± 3.27E-5 | 1.26 | 0.211 |
| | | | Gender | -2.78E-2 ± 7.85E-2 | -0.36 | 0.725 |
| Left uncinate | 0.056 | 65 | Age | 5.82E-5 ± 3.42E-5 | 1.71 | 0.093 |
| | | | Gender | 1.77E-2 ± 7.57E-2 | -0.23 | 0.816 |
| Right uncinate | 0.050 | 66 | Age | 5.48E-5 ± 3.10E-5 | 1.77 | 0.082 |
| | | | Gender | 3.40E-2 ± 6.99E-2 | 0.49 | 0.629 |
| Splenium | 0.003 | 66 | Age | 4.78E-7 ± 4.59E-5 | 0.01 | 0.992 |
| | | | Gender | 4.53E-2 ± 0.13 | 0.36 | 0.724 |

**PC2: Axon Packing and Myelin**

| Region | R² (adj) | df | Predictor | Estimate ± SE | t | p |
|---|---|---|---|---|---|---|
| Left arcuate | 0.181 | 66 | Age | **1.85E-4 ± 5.00E-5** | **3.70** | **0.0004**\* |
| | | | Gender | -1.45E-2 ± 0.11 | -0.13 | 0.894 |
| Right arcuate | 0.178 | 66 | Age | **1.95E-4 ± 5.34E-5** | **3.66** | **0.0005**\* |
| | | | Gender | -1.78E-2 ± 0.11 | -0.15 | 0.878 |
| Left ILF | 0.108 | 66 | Age | **1.37E-4 ± 4.99E-5** | **2.75** | **0.0077**\* |
| | | | Gender | -3.08E-2 ± 0.11 | -0.28 | 0.783 |
| Right ILF | 0.129 | 66 | Age | **1.53E-4 ± 5.01E-5** | **3.05** | **0.0033**\* |
| | | | Gender | -5.75E-2 ± 0.11 | -0.50 | 0.617 |
| Left IFOF | 0.137 | 66 | Age | **1.23E-4 ± 3.82E-5** | **3.21** | **0.0021**\* |
| | | | Gender | -4.35E-2 ± 8.90E-2 | -0.49 | 0.627 |
| Right IFOF | 0.195 | 66 | Age | **1.83E-4 ± 4.81E-5** | **3.80** | **0.0032**\* |
| | | | Gender | -9.65E-2 ± 0.11 | -0.90 | 0.372 |
| Left uncinate | 0.074 | 66 | Age | -5.06E-5 ± 6.88E-5 | 1.38 | 0.173 |
| | | | Gender | -0.81 ± 0.42 | 1.57 | 0.124 |
| Right uncinate | 0.025 | 66 | Age | 2.30E-5 ± 5.41E-5 | 0.42 | 0.673 |
| | | | Gender | 0.13 ± 0.12 | 1.11 | 0.274 |
| Splenium | 0.077 | 66 | Age | **2.05E-4 ± 8.87E-5** | **2.31** | **0.024**\* |
| | | 66 | Gender | 7.22E-2 ± 0.21 | 0.35 | 0.731 |

**PC3: Axon Diameter**

| Region | R² (adj) | df | Predictor | Estimate ± SE | t | p |
|---|---|---|---|---|---|---|
| Left arcuate | 0.030 | 66 | Age | -7.78E-6 ± 6.93E-5 | -0.11 | 0.911 |
| | | | Gender | 0.20 ± 0.15 | 1.28 | 0.207 |
| Right arcuate | 0.025 | 66 | Age | -5.46E-5 ± 7.58E-5 | -0.72 | 0.474 |
| | | | Gender | 0.16 ± 0.16 | 1.00 | 0.324 |
| Left ILF | 0.055 | 66 | Age | 1.10E-4 ± 6.17E-5 | 1.79 | 0.080 |
| | | | Gender | 9.28E-2 ± 0.13 | 0.74 | 0.465 |

(*Continued*)

**Table 2.** (Continued)

| | | | | | | |
|---|---|---|---|---|---|---|
| Right ILF | 0.098 | 66 | Age | -1.39E-4 ± 1.19E-4 | -1.16 | 0.248 |
| | | | Gender | -1.48 ± 0.74 | -2.01 | 0.049 |
| | | | Age*Gender | 3.92E-4 ± 1.68E-4 | 2.34 | 0.023 |
| Left IFOF | 0.029 | 66 | Age | 5.87E-5 ± 6.58E-5 | 0.89 | 0.377 |
| | | | Gender | 0.14 ± 0.14 | 0.98 | 0.334 |
| Right IFOF | 0.020 | 66 | Age | 2.39E-5 ± 6.76E-5 | 0.35 | 0.725 |
| | | | Gender | 0.14 ± 0.14 | 1.00 | 0.326 |
| Left uncinate | 0.096 | 66 | Age | 1.56E-4 ± 7.70E-5 | 2.02 | 0.047 |
| | | | Gender | 0.26 ± 0.16 | 1.68 | 0.097 |
| Right uncinate | 0.064 | 66 | Age | 6.92E-5 ± 6.77E-5 | 1.02 | 0.310 |
| | | | Gender | 0.25 ± 0.14 | 1.85 | 0.069 |
| Splenium | 0.040 | 66 | Age | -9.25E-5 ± 6.68E-5 | -1.39 | 0.172 |
| | | 66 | Gender | 0.11 ± 0.14 | 0.81 | 0.423 |

Significant effects that survive multiple comparisons are bolded and marked by an asterisk.

(reflecting a combination of increasing FA, MD, and AD and/or decreasing ODI). A similar relationship was observed in the right arcuate fasciculus but this finding did not survive multiple comparisons corrections. Positive relationships between PC2 and age were observed in the bilateral arcuate (L: t = 3.70, p < 0.001; R: t = 3.66, p < 0.001), inferior longitudinal fasciculus (L: t = 2.75, p = 0.007; R: t = 3.05, p = 0.003), inferior fronto-occipital fasciculus (L: t = 3.21, p = 0.002; R: t = 3.80, p = 0.003), and splenium (t = 2.31, p = 0.024). Increases in PC2 suggest increased axon packing and myelin with age (reflecting a combination of increases in FA and NDI, and/or decreases in MD and RD). The gender main effect (t = -2.01, p = 0.049) and the age*gender interaction were significant for PC3 in the right inferior longitudinal fasciculus, but neither survived multiple comparisons corrections. Scatterplots in Fig 5 illustrate relationships between PC1, PC2 and age.

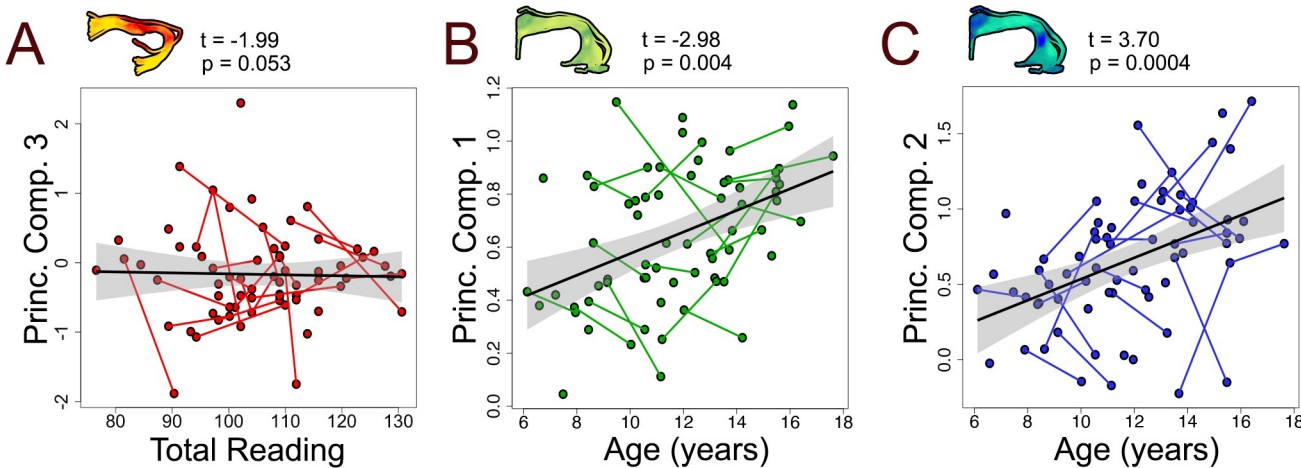

**Fig 5. Scatterplots visualizing relationships between principal component 3 (PC3) and Total Reading in the left uncinate fasciculus (A), PC1 and age in the left uncinate (B) and PC2 and age in the left uncinate (C).** Principal components are shown in an example tract for each relationship. Increases in PC1 indicate increased diffusion along a primary axis, while increases in PC2 indicate increased myelin and axon packing, thus relationships depicted in panels A and B could potentially reflect axonal maturation. No significant links between principal components and Total Reading were observed. The relationship between PC3 and Total Reading in the left uncinate was closest to our significance threshold.

**Table 3. Bayes factors assessing the likelihood of the null hypothesis condition (no relationship between Total Reading scores and model components) versus the likelihood of the model condition (relationships between included components and Total Reading).**

| READING MODELS | | |
|---|---|---|
| **Region** | **Components** | **Bayes Factor** |
| Left arcuate | PC1 + PC2 + PC3 + Age | 9.43 |
| Right arcuate | PC1 + PC2 + PC3 + Age | 8.93 |
| Left ILF | PC1 + PC2 + PC3 + Age | 47.62 |
| Right ILF | PC1 + PC2 + PC3 + Age | 20.83 |
| Left IFOF | PC1 + PC2 + PC3 + Age | 11.76 |
| Right IFOF | PC1 + PC2 + PC3 + Age | 9.35 |
| Left uncinate | PC1 + PC2 + PC3 + Age | 19.23 |
| Right uncinate | PC1 + PC2 + PC3 + Age | 19.23 |
| Splenium | PC1 + PC2 + PC3 + Age | 10.42 |

A Bayes factor of 3—indicating our sample data is 3 times more likely to be explained by the null condition than the model condition—or greater provides evidence for the null condition.

## 3.3 Bayes factor analysis

Bayes factors analysis was conducted to evaluate Total Reading mixed effects regression models. Results from this analysis are summarized in Table 3. Bayes factors including all principal components and age as covariates of Total Reading were greater than 3 in all regions, indicating evidence for the null hypothesis.

## Discussion

We applied principal component analysis in a multimodal dataset including highly specific measures of myelin, axon packing, and fiber coherence to investigate white matter development and links to reading. PCA identified three principal components that explained a large proportion of variance (79.5%) in our dataset, and represented tissue complexity (axon coherence), diffusion restriction (axonal packing and myelination), and axon diameter. The interpretation of principal components was based upon common sensitivities shared by the measures in each component and previous literature. The sensitivity of each individual metric included in PCA has been histologically validated [20, 28–33], suggesting that the interpretations presented here are biologically meaningful. PC1 explained the largest amount of variance (37.5%). With significant contributions from FA, MD, AD, and ODI, PC1 probed diffusion anisotropy and was driven by axon integrity and coherence. PC2 explained 23.0% of variance and reflects myelin and axonal packing, as shown by heavy loadings of FA, MD, RD, and NDI. Finally, PC3 explained 19.0% of variance and was driven by $VF_m$ and g-ratio. PC3 likely corresponds to axon diameter, as principal components are expected to be orthogonal and PC2 contains several myelin-sensitive measures. Studies employing PCA with white matter imaging measures have identified similar principal components related to diffusion anisotropy and overall diffusivity [34, 57]. Our PCA expands upon previous findings by including non-diffusion measures from magnetization transfer and relaxometry. This allowed our multimodal PCA to identify a novel third component related to axon diameter.

Shared information between white matter imaging metrics resulted in measures loading onto multiple principal components, in particular FA and MD. This was addressed in multiple ways. First, in the case of highly correlated variables, redundant variables (MTR) were

removed from PCA analysis. Next, varimax rotation minimized loading of a variable onto multiple principal components, and helped emphasize the differences between resultant principal components. Finally, re-running PCA without FA and MD resulted in a similar set of principal components accounting for 77.3% of variance and reinforcing our interpretation of the full model results. $PC1_B$ accounted for 36.6% of variance and was analogous to PC2 from the full model, with loadings from RD and NDI, along with qihMT which did not appear in the full model. $PC2_B$ accounted for 22.7% of variance was driven by $VF_m$ and g-ratio, similar to PC3. Finally, $PC3_B$ accounted for 18.0% of variance and had loadings from AD and ODI, similar to PC1. Principal component analysis with varimax rotation is shown to be an effective way to collapse white matter imaging metrics into powerful, interpretable measures. FA and MD were retained here to maintain power, though future studies may want to consider removal of broadly sensitive metrics such as FA and MD to improve specificity of resultant principal components.

Principal components were not significantly related to Total Reading scores in any investigated region. Bayes factors suggested the null hypothesis was substantially more likely than the alternative hypothesis in all regions. No significant relationships were identified in follow-up mixed effects models including principal components, age and scores from subtests included in the Total Reading composite score. Further, no significant correlations between initial measures and Total Reading scores were significant following multiple comparisons corrections. These findings suggest that gross relationships between white matter structural features and Total Reading ability are absent in typically developing children and adolescents, who tended to be skilled readers in our sample. Expanding this analysis to a larger age range or a population with reading difficulties may provide a larger effect to assess, and further insight into the role of white matter in reading.

Despite a lack of broad relationships between key white matter features and reading, some findings here hint that more specific relationships may be present in our sample. P-values < 0.1 suggest a larger sample may find significant relationships between PC2 or PC3 and Total Reading in the left IFOF and left uncinate, respectively. Left hemisphere ventral white matter supports reading processing in skilled readers, and left inferior frontal regions have been consistently highlighted as related to reading skill in previous studies [3, 6, 8–10]. Additionally, qihMT was correlated with Total Reading ability in the bilateral arcuate fasciculus and ILF, the right IFOF and right uncinate fasciculus, and was trend level in the left IFOF (see S1 Table), though these findings did not survive multiple comparison corrections. Interestingly, qihMT was not significantly related to Total Reading in either the left IFOF or uncinate fasciculus, where trend level relationships with principal components were found. Trend level relationships between PC2, PC3, or qihMT and Total Reading provide some evidence for a link between axon diameter and myelin and reading. However, these relationships must be investigated and confirmed by future studies.

Links between principal components and age were identified throughout the brain. Relationships between PC2 and age were most prominent, found in all tracts except the uncinate fasciculus, and are visualized as scatterplots in Fig 5. Age-related trends tended to be similar between left and right hemispheres, suggesting that at the macro-scale, brain development is similar between hemispheres. This is in contrast to investigations of individual microstructural features, where increases in $VF_m$ were shown to be largely left-lateralized during adolescence [58]. PC2 findings may be driven by NDI, as NDI has been previously shown to be age-sensitive and increases bilaterally throughout adolescence [58–60]. One relationship between PC1 and age remained in the left arcuate following multiple comparisons. While axon coherence tends to be stable across adolescence [61–63], we show that changes may still be ongoing in some regions. Gender was related to PC3 in the right inferior longitudinal fasciculus such that

males had higher values than females. Higher PC3 values reflect higher $VF_m$ and lower g-ratio values, thus the development of the right inferior longitudinal fasciculus may be further along in males. Studies of sex effects on white matter development have produced mixed results, suggesting either absence of or minor developmental effects during childhood and adolescence (for review see [64]), but large longitudinal studies remain necessary to effectively assess sex and gender effects across development.

This study has several limitations. First, inclusion of broadly sensitive measures such as FA and MD decreased clarity in interpretation of our principal components. We included these metrics to provide a baseline for future work applying principal component analysis in white matter, and to better connect to previous work. Future investigators should seek to refine their set of included metrics and exclude generally sensitive measures which may mask loadings of other, more specific metrics. Second, not all participants provided longitudinal data, and younger participants contributed fewer longitudinal data points than older participants. Future studies with more longitudinal data may be better able to elucidate relationships between components of white matter structure and age or reading across development. Finally, although the metrics applied here have been histologically validated, none are truly specific to any microstructural feature. Principal component analysis helps to address these sensitivities by focusing on information that is shared between measures, but our interpretation is still complicated by the multiple factors which affect each imaging metric.

## Conclusions

Here, we combined multimodal imaging techniques to assess microstructure in reading-related white matter tracts. Principal component analysis revealed three key features of white matter microstructure that explained 79.5% of variance in our dataset. Principal components were related to tissue complexity, axon packing and myelin, and axon diameter. No significant relationships were observed between principal components and Total Reading scores, suggesting gross relationships between white matter structural features and reading are not present in typical children and adolescents. Some trend level results suggest minor roles for axon diameter and myelin in reading ability, but these findings must be confirmed by further research. Principal components were sensitive to age effects, consistent with previous studies. PCA is an effective tool to preserve power and exploit shared variance between imaging metrics. Resultant principal components are age-sensitive have expanded our understanding of links between white matter and reading. This study provides an important initial description of PCA in a multimodal set of white matter imaging metrics, and will serve as an important baseline for future studies investigating white matter in development or cognitive disorders.

## Supporting information

**S1 Table. Correlations between measures included in the final PCA model and Total Reading in all investigated regions.** No correlations remained significant after correction for multiple comparisons.
(DOCX)

## Acknowledgments

The authors wish to thank the participants and families who participated in this study, without whom this work would not have been possible.

## Author Contributions

**Conceptualization:** Bryce L. Geeraert, Maxime Chamberland, Catherine Lebel.

**Data curation:** Bryce L. Geeraert, Catherine Lebel.

**Formal analysis:** Bryce L. Geeraert.

**Funding acquisition:** Bryce L. Geeraert, Catherine Lebel.

**Investigation:** Bryce L. Geeraert, Catherine Lebel.

**Methodology:** Bryce L. Geeraert, Maxime Chamberland, R. Marc Lebel, Catherine Lebel.

**Project administration:** Bryce L. Geeraert, Catherine Lebel.

**Resources:** Bryce L. Geeraert, R. Marc Lebel, Catherine Lebel.

**Software:** Bryce L. Geeraert, Maxime Chamberland, Catherine Lebel.

**Supervision:** Bryce L. Geeraert, Catherine Lebel.

**Validation:** Bryce L. Geeraert, Catherine Lebel.

**Visualization:** Bryce L. Geeraert, Catherine Lebel.

**Writing – original draft:** Bryce L. Geeraert.

**Writing – review & editing:** Bryce L. Geeraert, Maxime Chamberland, Catherine Lebel.

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
