## [Decision Letter · Decision Letter 0]

29 May 2020

PONE-D-20-12553

Multimodal principal component analysis to identify major features of white matter structure and links to reading

PLOS ONE

Dear Dr. Geeraert,

Thank you for submitting your manuscript to PLOS ONE. After careful consideration, we feel that it has merit but does not fully meet PLOS ONE’s publication criteria as it currently stands. Therefore, we invite you to submit a revised version of the manuscript that addresses the points raised during the review process.

We look forward to receiving your revised manuscript.

Kind regards,

Pew-Thian Yap

Academic Editor

PLOS ONE

Journal Requirements:

'I have read the journal's policy and the authors of this manuscript have the following

competing interests: Author RML is an employee of GE Healthcare.' 

We note that one or more of the authors are employed by a commercial company: GE Healthcare.

Additional Editor Comments (if provided):

Reviewers' comments:

Reviewer's Responses to Questions

**Comments to the Author**

1. Is the manuscript technically sound, and do the data support the conclusions?

Reviewer #1: Partly

Reviewer #2: Yes

2. Has the statistical analysis been performed appropriately and rigorously? 

Reviewer #1: No

Reviewer #2: Yes

3. Have the authors made all data underlying the findings in their manuscript fully available?

Reviewer #1: Yes

Reviewer #2: Yes

4. Is the manuscript presented in an intelligible fashion and written in standard English?

Reviewer #1: Yes

Reviewer #2: Yes

5. Review Comments to the Author

Reviewer #1: The authors proposed to study WM structures related to human reading function. Multiple dMRI derived parameters were extracted to quantify the microstructural properties of the WM. I have following comments

1. The introduction of the paper and the purpose of doing such a study is clearly stated, but it gets hard to understand when describing the methods, in particular the Statistical Analysis section. The section combines not only statistics but also the important part of the PCA analysis. I would suggest separating the PCA out in an individual section.

2. In addition, the PCA analysis should be introduced in more understandable way. From reading Section 2.5, a reader would not get a sense what the analysis is supposed to achieve. Also, there are many processing steps, such as varmax, lmer, and removal of correlated variables. It gets a little clear after reading the results, where there is information about the three components, removal of MTR. The reviewer thinks there is a need to significantly improve the clarity.

3. The result section uses AF as example for illustration of the 2 PCs. Can the authors provide visualization of other tracts?

4. There is a need to clarify why age is related. This may be related to the cohort under study?

5. Section 2.2: Please clarify the usage of each acquired imaging modality. Please clarify “14:24 minutes” and “SPGR”.

6. P6: Lmax - > L_max

7. P8: what are 20 segments? I guess 9 tracts mean 4 bilateral tracts plus one commissural tract.

8. P 8: Please further clarify the usage of KMO. Sampling of what aspects of the results?

9. P10: Are the two lines on the top equations? What is the purpose of them?

10. P11: Not sure how the 3 PC correspond to tissue complexity, myelin and axon packing, and axonal diameter.

11. The result seems to indicate the selected tracts and microstructural measures are more likely to be related to age, but not reading. This make the findings confusing compared to what the authors have proposed.

12. The Conclusion section seems redundant. Most information has been included in discussion.

13. Instead of discussing the techniques can enable different kinds of studies, it might be a good idea to discuss potential limitations.

Reviewer #2: The manuscript presented the association between white matter structure and reading ability in children from 6-16 years of age. Overall, the idea using PCA for multi metrics (from multiple modality) analysis is interesting. This could pave the way to interpret multi-dimensional data. The manuscript is well-written with exhaustive statistical analysis. There are some concerns needed to be addressed and clarified.

1. In 2.2 "Two diffusion-weighted datasets were acquired ...". Is this two different datasets of the same subjects or a multi-shell dataset? What is the consensus of scanning twice instead of just scan each subject once to avoid registration between different b-shells?

2. A figure summarizing the processing pipeline is needed. For example, it is unclear that the authors stated "[DTI] ... registration to skull-stripped T1 images ..." and later " [all measure maps] ... were registered to b = 900 s/mm2 FA maps ...". Is there two steps of registration, the first one is to register dMRI to T1 space and the second one is to register other modalities to dMRI space (in T1 space at that time) ?A pipeline figure would make the manuscript more intelligible.

3. Why REKINDLE model was chosen? If only the b=900 shell was use, a standard DTI dtifit from FSL is sufficient. REKINDLE was introduced for DKI. Given the authors have both b=900 and b=2000 shells and want to use REKINDLE, perhaps adding kurtosis properties in the analysis could be interesting.

4. It is unclear which tractography algorithm was used. "Whole brain tractography was performed using constrained spherical deconvolution [40]" and then "Next, semiautomated tractography [41] was performed". Are there 2 tractography steps? Also, given 2-shell data, multi-shell multi-tissue CSD could be better than simple CSD. In addition, [41] presented a semiautomated way to segment the tracts and used deterministic tractography: "A template was created based on 20 scans of one 25-year-old male. The images from these 20 scans were normalized to each other using an affine transformation and averaged to create the template. Non-diffusion-weighted images (b = 0 s/mm2) were registered to the template using an affine transformation followed by tensor reorientation. For each tract, seeding, target, and exclusion regions were selected manually on the template color map and automatically copied to each normalized brain. All voxels within the seeding region were used as seed points for fiber tracking for each of the 202 subjects, and the target and exclusion regions served to include or exclude fibers passing through specific areas. Fiber tracking was performed in ExploreDTI, software developed by one of the authors (A.L.), using a deterministic streamline method. FA thresholds were set to 0.25 to initiate and continue tracking, while the angle threshold was set to 60° for the uncinate fasciculus and the superior longitudinal fasciculus and 30° for all other tracts." Did the authors use the exact implementation? If so, what was the template, and how was deterministic tractography set?

5. What is the rationale behind "While removing FA and MD and running a reduced PCA model aided in interpretation of our principal components, mixed effects models regressions and Bayes factor analyses were conducted with the full PCA model including FA and MD." If removing FA and MD aids the interpretation, why the subsequent analyses were not performed with PCB?

6. What is the rationale in naming PC1 and tissue complexity and PC2 ad myelin and axonal packing? For the full model, PC1 consists of FA, MD, AD, and ODI, PC2 consists of FA, MD, RD, and NDI. Note that MD is the weighted average of AD and RD. So the only differences between PC1 and PC2 is ODI versus NDI. ODI is, however, does not represent how complex a voxel could be (number of compartments) but just how dispersed the fibers in a voxel. NDI could indicate axonal packing but it does not represent myelin content. A good way to represent myelin content is to use a T1/T2 ratio.

7. I'm more interested in the reduced model than the full model. MD is just a linear combination of AD and RD. Using MD with AD and RD could be redundant.

8. From Fig.2 and as the authors stated, the trend of some properties is very similar, which might not be useful. I would suggest adding some microstructure properties, such as the multi-compartment spherical mean technique (Kaden et al.) which is suitable for 2-shell data.

9. It is not clear how the longitudinal analysis for 22 subjects with re-scans after 2 years was carried out (or not)?

6. PLOS authors have the option to publish the peer review history of their article (what does this mean?). If published, this will include your full peer review and any attached files.

Reviewer #1: No

Reviewer #2: No

---

## [Author Response · Author response to Decision Letter 0]

15 Jul 2020

EDITOR COMMENTS / JOURNAL REQUIREMENTS:

- We have adjusted the format of our manuscript, supporting information, and figures to comply with PLOS ONE’s style requirements. Thank you for the links to these formatting documents.

- Agreed. The data we used in this analysis is publically available at FigShare: https://doi.org/10.6084/m9.figshare.12649388

'I have read the journal's policy and the authors of this manuscript have the following

competing interests: Author RML is an employee of GE Healthcare.' 

We note that one or more of the authors are employed by a commercial company: GE Healthcare.

- We have included updated conflicting interests and funding statements in our new cover letter.

REVIEWERS’ COMMENTS:

Reviewer #1: The authors proposed to study WM structures related to human reading function. Multiple dMRI derived parameters were extracted to quantify the microstructural properties of the WM. I have following comments

1. The introduction of the paper and the purpose of doing such a study is clearly stated, but it gets hard to understand when describing the methods, in particular the Statistical Analysis section. The section combines not only statistics but also the important part of the PCA analysis. I would suggest separating the PCA out in an individual section.

2. In addition, the PCA analysis should be introduced in more understandable way. From reading Section 2.5, a reader would not get a sense what the analysis is supposed to achieve. Also, there are many processing steps, such as varmax, lmer, and removal of correlated variables. It gets a little clear after reading the results, where there is information about the three components, removal of MTR. The reviewer thinks there is a need to significantly improve the clarity.

 In order to clarify the purpose and application of principal component analysis, we have made a few changes to our manuscript:

- First, we have reworked the final paragraph of our introduction to explicitly mention the issue of redundancy between metrics with shared sensitivities in univariate statistics, and note that principal component analysis takes advantage of these shared sensitivities.

- Second, we have created separate sections for principal component analysis and statistical analysis in the Methods chapter as suggested. Section 2.5 now describes principal component analysis in more detail, and more explicitly describes the sequence and purpose of each step included in our principal component analysis, including notes on the processes mentioned in comment 2:

o Varimax: “Varimax rotation was applied on retained principal components via the varimax() function to maximize differences in the loading of principal components onto input metrics and improve interpretation of component sensitivities.”

o Removal of correlated variables: “Following PCA, input variable contributions to principal components along with correlations between variables within along-tract data were inspected to identify redundancy between variables. In the case of highly collinear measures (moderately to highly correlated (|r| > 0.6) and contributed to PCA outputs similarly), the variable with highest correlations to all other input measures was removed to improve stability of PCA computations [1] and PCA was recomputed.”

o KMO (related to comment 8): “A Kaiser-Meyer Olkin (KMO) test was conducted via the KMO() function to assess correlation between input measures and indicate the suitability of our measure set for PCA, with a value of above 0.5 indicating suitability [2].”

- Section 2.6 describes all statistical analyses carried out after PCA, includes a reference for the lmer tool used for linear mixed effects models, and notes that lmer was applied in order to include available longitudinal data:

o Lmer: “Linear mixed effects models were computed via lmer() to investigate relationships between principal components with Total Reading and age in each tract. Age models included age, gender, an age*gender interaction, and a random intercept per subject, to accommodate for data from subjects who had participated twice.”

3. The result section uses AF as example for illustration of the 2 PCs. Can the authors provide visualization of other tracts?

- Principal components are visualized in the arcuate fasciculus in Figure 4 (previously figure 3) as an example of how our latent principal components vary across tracts, similarly to figure 3 (previously figure 2). Our regression analyses were carried out using mean principal components in each tract rather than along-tract measures, so we chose not to produce visualizations of principal components in all tracts to avoid causing readers to assume that regressions were conducted using along-tract data. 

- We revised Figure 5 (previously figure 4) to better represent our regression analyses. This included changing our example tracts to heat maps of principal components in the relevant tract, and connecting data points which belong to the same participant. This should further help to clarify how principal components were used in later analyses.

4. There is a need to clarify why age is related. This may be related to the cohort under study?

- Indeed, the age relationships reflect the fact that this is a pediatric sample. Age was included in the model a necessary covariate, since the metrics included in the PCA are related to age, and we opted to interpret these results to both to take advantage of our pediatric cohort and to build upon previous work by Chamberland et al. [3] showing that principal components are biologically relevant and age-sensitive. We have reworked the final paragraph of our introduction to better explain why we looked at both reading and age. Key phrases include:

o “A framework using PCA for dimensionality reduction in white matter has been recently described [3] and resultant components were linked to age, suggesting developmental sensitivity.”

o “We aimed to investigate links between resultant principal components with both age and reading to describe development of key microstructural features and how these features underlie reading.”

o “Furthermore, we expected that these components would be linked to age and reading proficiency in reading-related tracts, such that indications of more myelin, axonal packing, and fiber coherence would be observed with increasing age and would relate to better reading performance.”

- Additionally, we have added a final sentence to the abstract to more clearly connect the two investigations:

o “Our findings suggest major features of white matter undergo adolescent development, but these changes not linked to reading during this period.”

5. Section 2.2: Please clarify the usage of each acquired imaging modality. Please clarify “14:24 minutes” and “SPGR”.

- “14:24 minutes” referred to 14 minutes and 24 seconds. We have added both units to the times (“min:sec”), replaced usage of “scan duration” with scan time, and placed all scan times at the end of descriptions of each MRI sequence to standardize presentation of this information.

- We have adjusted wording to introduce SPGR as “spoiled gradient (SPGR)” and for IR-SPGR, introducing as “inversion recovery SPGR (IR-SPGR)”.

- The usage of each imaging modality described in section 2.2 is provided in section 2.3 “Image Processing”. We have made several modifications to ensure that usage of each imaging sequence is clearly described, and have re-tooled the first paragraph on DTI data processing to clarify how b=900 and b=2000s/mm2 datasets were processed. Furthermore, a new figure has been produced to provide a visual depiction of our processing pipeline. This figure is presented as Figure 2 in our updated manuscript.

6. P6: Lmax - > L_max

- Thank you for the correction, we have adjusted to “L_max” on the bottom of page 6.

7. P8: what are 20 segments? I guess 9 tracts mean 4 bilateral tracts plus one commissural tract.

- You are correct. The tracts of interest are listed at the end of page 7 and depicted in Figure 1, which notes that tracts were investigated bilaterally in the figure caption.

- We have clarified the meaning of “20 segments” in two ways. First, at the end of section 2.3 when discussing extraction of measure means in our 9 tracts of interest, we have rephrased the final sentence to: “Additionally, along-tract analysis was performed in ExploreDTI [4, 5], to sample all ten measures at twenty equidistant points along each tract.” Next, in section 2.5, we have described data included in principal component analysis as “(10 measures x 9 tracts of interest x 20 points sampled along each tract)” to again note that we performed PCA on data obtained by along-tract analysis.

8. P 8: Please further clarify the usage of KMO. Sampling of what aspects of the results?

- The Kaiser-Meyer-Olkin test assesses how much variance in a set of data may be explained by underlying factors. The KMO test value ranges from 0 to 1, with higher values indicating greater shared information between measures, and higher suitability for factor analysis.

- To clarify the purpose of the kMO test, section 2.5 now includes: “A Kaiser-Meyer-Olkin (KMO) test was conducted via the KMO() function to assess correlation between input measures and indicate the suitability of our measure set for PCA, with a value of above 0.5 indicating suitability [2].”

- The KMO test is also discussed further in the new limitations section at the end of our discussion, because our data neared the threshold for ill-suitability to PCA.

9. P10: Are the two lines on the top equations? What is the purpose of them?

- This may have been a formatting error in our manuscript. All equations in the manuscript (six total: four equations in the body, one equation each in the captions of Table 1 and Table 2) have been replaced with equations formatted by the Equation tool in Word to avoid any unintentional formatting issues.

10. P11: Not sure how the 3 PC correspond to tissue complexity, myelin and axon packing, and axonal diameter.

- Thank you for the feedback. It took quite a bit of careful consideration to interpret these principal components, and we would like to make sure that our interpretation is clearly presented and justified within this paper. This is primarily described in the discussion section, but we have added a sentence to section 3.1 of our results here: “Interpretation of principal components was carried out by evaluating the common microstructural sensitivities of each measure, and by comparison to previous PCA analyses in white matter [34, 56].”

- Our interpretation of principal components is elaborated upon at the beginning of our discussion (paragraphs 2 and 3). While too much to reiterate here, one key sentence describes the interpretation process and refers readers to previous literature outlining the microstructural sensitivities of each measure, should they wish to learn more:

o “The interpretation of principal components was based upon common sensitivities shared by the measures upon which each component loaded. Furthermore, as each the sensitivity of each metric included in PCA has been histologically validated, the interpretations presented here are biologically meaningful [20, 28-33].”

11. The result seems to indicate the selected tracts and microstructural measures are more likely to be related to age, but not reading. This make the findings confusing compared to what the authors have proposed.

- We agree. The absence of reading findings was unexpected, especially because previous DTI-based literature routinely reports correlations with reading in left temporoparietal and ventral tracts such as those we investigated. Additionally, correlations between individual measures included in the supplementary table hint that more specific relationships may be present, particularly between qihMT and reading, although these findings do not survive multiple comparisons. However, precisely because we did not see the expected trend, we have an opportunity here to add completely new information to our understanding of links between white matter and reading.

- We propose that our absence of findings is due in part to the broad lens through which we have investigated white matter & reading (linking 3 major features of white matter structure to a composite score of overall reading ability), rather than searching for more specific relationships between individual microstructural measures and components of reading such as phonological processing, vocabulary, and more. Our findings show that at the broad level in our typically-developing sample, white matter is not an important determinant of reading ability. However, previous findings and our own supplemental analysis suggests that smaller factors may still interact, such as myelin in tracts (qihMT) and reading. This relationship can likely be broken down to smaller factors in reading ability such as phonological processing.

- Additionally, we have assessed a mature sample of typically-developing, skilled readers. Comparison to a group with reading difficulties or dyslexia would provide more variance in reading scores, and may uncover relationships between white matter microstructure and reading that we were unable to observe here.

- Despite the counter-intuitive results, we maintain our original hypothesis in the introduction in order to preserve the description of our scientific process, and comment upon the unexpected findings in both the abstract and discussion, with the newly-added limitations section for extra context.

12. The Conclusion section seems redundant. Most information has been included in discussion.

- The purpose of our conclusions section was to briefly summarize our discussion and restate the main themes. We have reworked this section to significantly shorten the text. The conclusions now follows our new limitations section, helping give it a more defined role of wrapping up the manuscript.

13. Instead of discussing the techniques can enable different kinds of studies, it might be a good idea to discuss potential limitations.

- Yes, excellent point. We have added a new paragraph to discuss the limitations of this study at the end of our discussion. This section discusses the observed moderate KMO test value (indicating only moderate suitability for factor analysis), incomplete longitudinal data, and additional sensitivities present for each applied imaging measure. We opted to leave our discussion of limitations in our analysis of reading to the relevant discussion section on pages 20 and 21, as several pieces of evidence introduced in this section provide important context.

Reviewer #2: The manuscript presented the association between white matter structure and reading ability in children from 6-16 years of age. Overall, the idea using PCA for multi metrics (from multiple modality) analysis is interesting. This could pave the way to interpret multi-dimensional data. The manuscript is well-written with exhaustive statistical analysis. There are some concerns needed to be addressed and clarified.

1. In 2.2 "Two diffusion-weighted datasets were acquired ...". Is this two different datasets of the same subjects or a multi-shell dataset? What is the consensus of scanning twice instead of just scan each subject once to avoid registration between different b-shells?

- We have adjusted this sentence to read “two diffusion-weighted datasets were sequentially acquired…” to clarify that b=900 and b=2000 s/mm2 datasets were acquired separately. Aside from diffusion-weighting, the parameters of these scans were identical. Therefore, registration between b-shells was minor and can be considered similar to the coregistration between volumes commonly used to correct for motion artifacts.

- As we were scanning kids, separating b=900 and b=2000 s/mm2 scans allowed us to re-run either sequence in the case that the subject moved their head during acquisition, producing unusable volumes of data. As we were scanning children as young as 6 years of age, subject motion occasionally resulted in multiple scan sequence repeats to acquire good quality data.

2. A figure summarizing the processing pipeline is needed. For example, it is unclear that the authors stated "[DTI] ... registration to skull-stripped T1 images ..." and later " [all measure maps] ... were registered to b = 900 s/mm2 FA maps ...". Is there two steps of registration, the first one is to register dMRI to T1 space and the second one is to register other modalities to dMRI space (in T1 space at that time) ?A pipeline figure would make the manuscript more intelligible.

- Thank you for the feedback. We have developed a processing pipeline figure and included it in the manuscript as Figure 2. All other figure numbers have been adjusted accordingly.

- To your question here: You are correct that there were two registration steps. During dMRI processing in ExploreDTI, b = 900 s/mm2 data was registered to subject T1 images. This is done to correct for EPI distortions in the images. NODDI, mcDESPOT, and ihMT measures were produced in their native spaces then registered to dMRI images in T1 space, resulting in all measure maps in T1 space.

3. Why REKINDLE model was chosen? If only the b=900 shell was use, a standard DTI dtifit from FSL is sufficient. REKINDLE was introduced for DKI. Given the authors have both b=900 and b=2000 shells and want to use REKINDLE, perhaps adding kurtosis properties in the analysis could be interesting.

- The REKINDLE model was recommended to us by the creators of ExploreDTI as an appropriate way to account for outlier data when computing the tensor. The initial publication describing the REKINDLE method notes that while developed for DKI, it is applicable to DTI [6].

- Many different white matter imaging measures are available which may provide additional nuances to build upon the findings presented here. This includes diffusion kurtosis imaging, fixel-based analyses, and many more. We chose a set of imaging measures while simultaneously balancing high reliability in our age range, capturing a broad range of microstructural features, and reducing redundancy between our techniques.

4. It is unclear which tractography algorithm was used. "Whole brain tractography was performed using constrained spherical deconvolution [40]" and then "Next, semiautomated tractography [41] was performed". Are there 2 tractography steps? Also, given 2-shell data, multi-shell multi-tissue CSD could be better than simple CSD. In addition, [41] presented a semiautomated way to segment the tracts and used deterministic tractography: "A template was created based on 20 scans of one 25-year-old male. The images from these 20 scans were normalized to each other using an affine transformation and averaged to create the template. Non-diffusion-weighted images (b = 0 s/mm2) were registered to the template using an affine transformation followed by tensor reorientation. For each tract, seeding, target, and exclusion regions were selected manually on the template color map and automatically copied to each normalized brain. All voxels within the seeding region were used as seed points for fiber tracking for each of the 202 subjects, and the target and exclusion regions served to include or exclude fibers passing through specific areas. Fiber tracking was performed in ExploreDTI, software developed by one of the authors (A.L.), using a deterministic streamline method. FA thresholds were set to 0.25 to initiate and continue tracking, while the angle threshold was set to 60° for the uncinate fasciculus and the superior longitudinal fasciculus and 30° for all other tracts." Did the authors use the exact implementation? If so, what was the template, and how was deterministic tractography set?

- Correct, whole brain tractography and semiautomated tract delineation are separate processing steps. Note that while the semiautomated tractography tool implemented in ExploreDTI was inspired by the work presented in [41], the tool has been adapted to work with any dataset (additional details can be found in the ExploreDTI manual: http://www.exploredti.com/manual/Manual_ExploreDTI.pdf). The processing steps you have quoted above match our own, with three key differences:

o One exemplar subject was chosen from our cohort. Thus our template scan was a single dataset, rather than an average of multiple scans.

o Registration was carried out based upon FA maps, rather than b0 images, as this is how the tools has been implemented in ExploreDTI and improves registration accuracy.

o Deterministic streamline tractography was carried out as described in section 2.3 of our Methods chapter.

- We have adjusted our description of whole brain and semiautomated tractography in section 2.3 to provide additional details regarding tractography parameters and clarify that these are two separate steps. We have also developed a processing pipeline figure, included in our revised manuscript as Figure 2 (all other figure numbers have been adjusted).

- Here is a step-by-step outline the semiautomated tractography process: 

o We chose an exemplar subject in the middle of our sample’s age range with high quality data. This choice is somewhat subjective, so in order to ensure we chose an appropriate subject we performed in-house testing to evaluate the feasibility of resultant tracts using different exemplar subjects, and looked for qualitative differences and quantitative differences in mean FA, MD, AD, and RD. Findings of this in-house testing were that datasets with no bad volumes in the DTI dataset produced similar results.

o Next, ROIs were manually drawn to segment all tracts of interest in this exemplar subject. ROIs were based on Wakana et al. [7] and manually dilated by 1mm to ensure no possible streamlines belonging to this tract were excluded. 

o Next, for each additional subject in our cohort, the semiautomated tractography algorithm registered the exemplar subject’s FA map to the subject FA map, then applied the same transformations to warp our manually drawn ROIs into the new subject’s space. 

o Next, our chosen tracts of interest were segmented from whole brain tractography files produced during previous processing (see the new Figure 2 panel E) to produce individualized tracts for each subject. 

o Finally, all subject tracts were visually inspected, rated, and trimmed of any spurious fibers that remained after semiautomated tractography.

- Regarding the use of multi-tissue CSD: We found that with our participant cohort and available data quality, CSD tractography using our b = 900 s/mm2 shell DTI dataset provided the best balance between anatomically plausible tractogram results, appropriate tract segmentations and spurious streamlines. The b = 2000 s/mm2 datasets are more sensitive to subject motion and thus were more likely to contain bad volumes. In order to keep tractography results as consistent as possible, we chose to perform tractography on data that was consistently high quality across our cohort.

5. What is the rationale behind "While removing FA and MD and running a reduced PCA model aided in interpretation of our principal components, mixed effects models regressions and Bayes factor analyses were conducted with the full PCA model including FA and MD." If removing FA and MD aids the interpretation, why the subsequent analyses were not performed with PCB?

- We have added a sentence to the end of section 3.1 (page 12) which reads: “Further statistical analyses were conducted using the full model to preserve power to detect effects with age and reading. Furthermore, a KMO test value of 0.43 for our reduced model indicates input variables may not share enough information for robust factor analysis.”

6. What is the rationale in naming PC1 and tissue complexity and PC2 ad myelin and axonal packing? For the full model, PC1 consists of FA, MD, AD, and ODI, PC2 consists of FA, MD, RD, and NDI. Note that MD is the weighted average of AD and RD. So the only differences between PC1 and PC2 is ODI versus NDI. ODI is, however, does not represent how complex a voxel could be (number of compartments) but just how dispersed the fibers in a voxel. NDI could indicate axonal packing but it does not represent myelin content. A good way to represent myelin content is to use a T1/T2 ratio.

- This was the subject of much discussion by the authors, as the remainder of the experiment hinged upon the sensitivities of our principal components. First, regarding the meaning of ‘tissue complexity’, here we refer to the complexity of fiber orientations in a voxel rather than the number of compartments. This is consistent with [3]. Second, we considered all factors known to influence each imaging metric, along with results from previous principal component analyses, to determine which factors most likely drove each principal component observed here. The arguments for our interpretation are as follows:

1. Principal component analysis produces orthogonal components (no shared variance between components), thus it is likely that the loadings of PC1 and PC2 onto FA and MD are driven by separate sensitivities within each measure. 

2. De Santis et al [8] performed a principal component analysis using 8 white matter imaging metrics and identified 3 principal components explaining 78% of variance in their data, similar to our own results. However, many input metrics appeared in multiple principal components, particularly diffusion metrics. This makes interpretation of their principal components difficult but shows that diffusion metrics can be expected to be generally sensitive and present in multiple principal components.

3. Chamberland et al found two principal components similar to our own. Their PC1 loaded onto fiber density, radial diffusivity, and restricted signal fraction similar to our PC2 loadings onto NDI, RD, and FA/MD, respectively. Further, their PC2 loaded onto number of fiber orientations, AD, and MD, similar to our PC1 loadings onto ODI, AD, and MD, respectively. We expected our results to be similar to the components described by Chamberland et al. due to our inclusion of six diffusion-based metrics, but we included metrics from non-diffusion-based MRI sequences, a key difference in our project which helped us to elaborate upon previous findings.

4. FA, RD, qihMT, VFm, and g-ratio in our dataset are all influenced by myelin, with qihMT and VFm being the most specific and histologically validated. Indeed, qihMT and VFm are generally more myelin-specific than T1/T2 ratio, which can be influenced by inflammation and has been described as only modestly related to myelin [9]. From our results, we argue that potential contributions from qihMT and VFm were precluded by diffusion measures, especially FA and RD. Removal of FA and MD from the analysis resulted in a loading of PC2 onto qihMT, further reinforcing our interpretation of PC2 as a myelin-sensitive component.

5. Finally, as VFm is a volume fraction of myelin, it is also influenced by axonal water content, resulting in a loading of PC3 onto VFm. G-ratio is also influenced by axonal water fraction alongside myelin water content, thus we interpret PC3 as sensitive to axonal water content.

7. I'm more interested in the reduced model than the full model. MD is just a linear combination of AD and RD. Using MD with AD and RD could be redundant.

- While DTI metrics are all derived from the same three eigenvalues, we included FA, MD, AD, and RD as each has unique sensitivities to add to our model. Additionally, as PCA is designed to produce orthogonal components and we checked for highly collinear measures which contributed to principal components in similar ways, we were confident that redundancy in input measures would be accounted for. This methodology resulted in the removal of MTR (redundant with qihMT), but MD, AD, and RD were all retained as they contributed unique information to latent principal components.

- Furthermore, previous PCA analyses in white matter [3, 8] have included FA, MD, AD, and RD, thus inclusion of these metrics in our own analyses helped to extend upon previous findings.

- Future PCA analyses may elect to remove FA and MD to take advantage of the higher specificity of other measures, but we retain FA and MD in the analysis presented here both to be consistent with our initial aims and hypotheses, and because we found a sub-threshold KMO test value for our reduced model.

8. From Fig.2 and as the authors stated, the trend of some properties is very similar, which might not be useful. I would suggest adding some microstructure properties, such as the multi-compartment spherical mean technique (Kaden et al.) which is suitable for 2-shell data.

- Indeed, several imaging metrics which shared variance were included in our study. There is a lack of previous research to base decisions upon which metrics to include and exclude in our analysis, making decisions of which metric to include and exclude based upon shared variance subjective. 

- One key reason to implement PCA in our analysis was to apply a method which could collapse shared variance, reducing issues with multiple comparisons introduced by redundancy between measures. Additionally, we followed the methods of Chamberland et al. in part because they describe multiple steps to visualize and evaluate redundancy between input measures. We have adapted our description of this process (page 9) to now read: “Following PCA, input variable contributions to principal components along with correlations between variables within along-tract data were visually and quantitatively inspected to identify redundancy between variables. In the case of highly collinear measures (moderately to highly correlated (|r| > 0.6) and contributed to PCA outputs similarly), the variable with highest correlations to all other input measures was removed to improve stability of PCA computations [1] and PCA was recomputed.”

- We include several imaging metrics in our study which offer improved microstructural sensitivity over DTI metrics. One key advantage of the spherical mean technique is its ability to probe intra- and extra-neurite diffusion compartments while factoring out the effect of fiber orientation dispersion. Here we applied neurite orientation dispersion and density imaging (NODDI), another advanced diffusion model suitable for 2-shell diffusion data, to produce similar, microstructurally sensitive metrics to the Kaden et al. technique. NODDI’s neurite density index (NDI) was used to assess axonal water content (related to axonal packing), while orientation dispersion is accounted for in the NODDI model is well via inclusion of the orientation dispersion index (ODI).

- Beyond NODDI metrics, we also include two metrics which have been histologically validated as myelin sensitive, quantitative ihMT (qihMT) and myelin volume fraction (VFm), and the g-ratio which is calculated from NDI and VFm and provides a useful perspective on the communication efficiency of fibers in a region.

9. It is not clear how the longitudinal analysis for 22 subjects with re-scans after 2 years was carried out (or not)?

- Longitudinal data was included in linear mixed-effects models regressions, which accounts for multiple time points per subject and incomplete longitudinal data, with subject as a random variable. We have refined our description of linear mixed-effects models in section 2.6 to read: “Linear mixed effects models were computed via lmer() [54] to investigate relationships between principal components with Total Reading and age in each tract. Age models included age, gender, an age*gender interaction, and a random intercept per subject, to account for repeated measures within subjects.”

- Additionally, we have connected data points from the same subjects on scatterplots in Figure 5 (previously figure 4) to provide a visual prompt for the inclusion of our longitudinal data.

 

REFERENCES

1. Garg A, Tai K. Comparison of statistical and machine learning methods in modelling of data with collinearity. IJMIC. 2013;18:295-312.

2. Dziuban CD, Shirkey EC. When is a correlation matrix appropriate for factor analysis? Some decision rules. Psychological Bulletin. 1974;81(6):358-61.

3. Chamberland M, Raven EP, Genc S, Duffy K, Descoteaux M, Parker GD, et al. Dimensionality reduction of diffusion MRI measures for improved tractometry of the human brain. NeuroImage. 2019;200:89-100. doi: 10.1016/j.neuroimage.2019.06.020. PubMed PMID: 31228638.

4. Colby JB, Soderberg L, Lebel C, Dinov ID, Thompson PM, Sowell ER. Along-tract statistics allow for enhanced tractography analysis. NeuroImage. 2012;59(4):3227-42. doi: 10.1016/j.neuroimage.2011.11.004. PubMed PMID: 22094644; PubMed Central PMCID: PMCPMC3288584.

5. Yeatman JD, Dougherty RF, Myall NJ, Wandell BA, Feldman HM. Tract profiles of white matter properties: automating fiber-tract quantification. PLoS One. 2012;7(11):e49790. doi: 10.1371/journal.pone.0049790. PubMed PMID: 23166771; PubMed Central PMCID: PMCPMC3498174.

6. Tax CM, Otte WM, Viergever MA, Dijkhuizen RM, Leemans A. REKINDLE: robust extraction of kurtosis INDices with linear estimation. Magnetic resonance in medicine. 2015;73(2):794-808. doi: 10.1002/mrm.25165. PubMed PMID: 24687400.

7. Wakana S, Jiang H, Nagae-Poetscher LM, van Zijl PC, Mori S. Fiber Tract-based Atlas of Human White Matter Anatomy. Radiology. 2004;230:77-87.

8. De Santis S, Drakesmith M, Bells S, Assaf Y, Jones DK. Why diffusion tensor MRI does well only some of the time: variance and covariance of white matter tissue microstructure attributes in the living human brain. NeuroImage. 2014;89:35-44. doi: 10.1016/j.neuroimage.2013.12.003. PubMed PMID: 24342225; PubMed Central PMCID: PMCPMC3988851.

9. Arshad M, Stanley JA, Raz N. Test-retest reliability and concurrent validity of in vivo myelin content indices: Myelin water fraction and calibrated T1 w/T2 w image ratio. Human brain mapping. 2017;38(4):1780-90. doi: 10.1002/hbm.23481. PubMed PMID: 28009069; PubMed Central PMCID: PMCPMC5342928.

---

## [Decision Letter · Decision Letter 1]

3 Aug 2020

Multimodal principal component analysis to identify major features of white matter structure and links to reading

PONE-D-20-12553R1

Dear Dr. Geeraert,

We’re pleased to inform you that your manuscript has been judged scientifically suitable for publication and will be formally accepted for publication once it meets all outstanding technical requirements.

Kind regards,

Pew-Thian Yap

Academic Editor

PLOS ONE

Additional Editor Comments (optional):

Reviewers' comments:

Reviewer's Responses to Questions

**Comments to the Author**

1. If the authors have adequately addressed your comments raised in a previous round of review and you feel that this manuscript is now acceptable for publication, you may indicate that here to bypass the “Comments to the Author” section, enter your conflict of interest statement in the “Confidential to Editor” section, and submit your "Accept" recommendation.

Reviewer #1: All comments have been addressed

Reviewer #2: All comments have been addressed

2. Is the manuscript technically sound, and do the data support the conclusions?

Reviewer #1: Yes

Reviewer #2: Yes

3. Has the statistical analysis been performed appropriately and rigorously? 

Reviewer #1: Yes

Reviewer #2: Yes

4. Have the authors made all data underlying the findings in their manuscript fully available?

Reviewer #1: Yes

Reviewer #2: Yes

5. Is the manuscript presented in an intelligible fashion and written in standard English?

Reviewer #1: Yes

Reviewer #2: Yes

6. Review Comments to the Author

Reviewer #1: The authors have addressed all my concerns. Thank you. I recommend the paper to be published in Plosone.

Reviewer #2: (No Response)

7. PLOS authors have the option to publish the peer review history of their article (what does this mean?). If published, this will include your full peer review and any attached files.

Reviewer #1: No

Reviewer #2: No

---

## [Editor Report · Acceptance letter]

5 Aug 2020

PONE-D-20-12553R1 

Multimodal principal component analysis to identify major features of white matter structure and links to reading 

Dear Dr. Geeraert:

I'm pleased to inform you that your manuscript has been deemed suitable for publication in PLOS ONE. Congratulations! Your manuscript is now with our production department. 

Kind regards, 

on behalf of

Dr. Pew-Thian Yap 

Academic Editor

PLOS ONE